# THSD7A Positivity Predicts Poor Survival and Is Linked to High FAK Expression and FGFR1-Wildtype in Female Patients with Squamous Cell Carcinoma of the Lung

**DOI:** 10.3390/ijms241310639

**Published:** 2023-06-26

**Authors:** Fidelis Andrea Flockerzi, Johannes Hohneck, Frank Langer, Rainer Maria Bohle, Phillip Rolf Stahl

**Affiliations:** 1Department of Pathology, Saarland University Medical Center, 66424 Homburg, Germany; 2Department of Thoracic and Cardiovascular Surgery, Saarland University Medical Center, 66424 Homburg, Germany

**Keywords:** THSD7A, FAK, LSCC

## Abstract

Lung cancer is the leading cause of cancer-related deaths in the western world, with squamous cell carcinoma being one of the most common histological subtypes. Prognostic and predictive markers are still largely missing for squamous cell carcinoma of the lung (LSCC). Several studies indicate that THSD7A might at least play a role in the prognosis of different tumors. FAK seems to play an important role in lung cancer and is discussed as a potential therapeutic target. In addition, there is evidence that FAK-dependent signaling pathways might be affected by THSD7A. For that reason, we investigated the role of THSD7A as a potential tumor marker in LSCC and whether THSD7A expression has an impact on the expression level of FAK. A total of 101 LSCCs were analyzed by immunohistochemistry using tissue microarrays. THSD7A positivity was associated with poor overall survival in female patients and showed a relation to high FAK expression in this subgroup. To our knowledge, we are the first to report these correlations in lung cancer. The results might be proof of the assumed activation of FAK-dependent signaling pathways by THSD7A and that as a membrane-associated protein, THSD7A might serve as a putative therapeutic target in LSCC.

## 1. Introduction

Lung cancer is one of the most common types of human cancer and the leading cause of cancer-related deaths in the western world [1]. Squamous cell carcinoma represents one of the most common histological subtypes of lung cancer. Several established predictive markers exist for adenocarcinomas of the lung, which enable clinicians to perform targeted therapy for this cancer subtype. Such promising prognostic and predictive markers are missing in squamous cell carcinoma of the lung (LSCC), with the exception of PD-L1 and rare mutations in Epidermal Growth Factor Receptor (EGFR) and rearrangements in Anaplastic Lymphoma Kinase (ALK) [2,3,4]. Regarding the prognosis of LSCC, which is still poor, novel predictive (molecular) markers and targets are desirable. 

In a recent study, we could show that positivity of Thrombospondin Type-1 Domain-Containing 7A (THSD7A) is associated with a high expression of Focal Adhesion Kinase (FAK) and with adverse clinicopathological parameters in prostate cancer [5]. THSD7A became known because of its involvement in the pathogenesis of membranous nephropathy (MN) [6,7,8]. There is also evidence of the participation of THSD7A in angiogenesis [9,10,11] and an association with obesity [12]. Furthermore, several studies indicate that THSD7A might at least play a role in the prognosis of different tumor types [13,14,15,16,17,18]. In addition to that, other groups report on THSD7A expression in different tumor types and a possible relationship between malignant diseases and MN [6,16,17,19,20,21,22,23,24,25]. 

There are a few studies that deal with THSD7A and lung cancer. However, the investigators concentrated rather on the coincidence of lung cancer and THSD7A-positive MN than on the role of THSD7A as a potential tumor marker in lung cancer [26,27,28] Data on the association between THSD7A expression and clinicopathological parameters in LSCC are missing so far.

There is evidence that Focal Adhesion Kinase (FAK) might be activated by THSD7A [5,10,11]. FAK is a protein tyrosine kinase that regulates cellular adhesion, motility, proliferation, and survival in various types of cells. FAK is overexpressed in several tumor types and is believed to play a role in tumor progression and metastasis [29,30,31,32]. For that reason, FAK is also discussed as a cancer target in various tumors [33,34,35,36,37]. In non-small cell lung cancer (NSCLC), FAK is reported to be overexpressed compared to normal lung tissue [38,39,40,41,42]. Furthermore, FAK seems to play a role in tumor progression and metastasis in NSCLC [43,44].

We conducted this study to investigate the role of THSD7A as a potential biomarker in LSCC and to see whether THSD7A positivity has an impact on the expression of FAK in its unphosphorylated form. For that, a previously described cohort of 101 tumors [41] was analyzed by immunohistochemistry (IHC) using tissue microarrays (TMAs).

In this former study, it was shown that the amplification of Fibroblast growth factor receptor 1 (FGFR1) is a frequent genetic alteration in LSCC with a mutation frequency of about 22%. FGFR1 gene amplification was associated with late tumor stages, and in female patients this alteration was linked to better overall survival [41].

Fibroblast growth factor receptors (FGFRs) are tyrosine kinase receptors regulating cell growth and cell proliferation through the stimulation of numerous intracellular signaling cascades. FGFRs are considered to be oncogenes, and alterations of FGFR are found in different tumor types. One of the most frequent alterations is FGFR1 gene amplification, notably occurring in ovarian cancer, head and neck cancer, urothelial cancer and lung cancer, with a frequency of 9 to 28% [45,46]. FGFR1 is considered to be a driver lesion in LSCC and even a potential biomarker for treatment of lung cancer [47]. ERK und FAK are considered to be two of the main depending intracellular pathways stimulated by FGFR1 in lung and ovarian cancer [48]. Furthermore, there is evidence that the FGFR1-depending activation of ERK and FAK affects epithelial-mesenchymal transition and leads to tumor growth and distant organ metastasis in lung cancer [49]. Several FGFR inhibitors have been developed in recent years. However, these agents showed only a low efficacy in clinical trials so far [50]. Hence, the prognostic and therapeutic value of FGFR1 gene amplification remains unclear. Based on the previously collected data on FGFR1 in LSCC, we were able to investigate a possible connection between FGFR1 with THSD7A and FAK, respectively.

## 2. Results

A total of 91 (90.1%) tumors were analyzable for THSD7A-IHC. A total of eight (8.8%) tumors showed at least a weak positivity with a membranous and/or cytoplasmic staining pattern. 

Twenty-one female patients were analyzable for THSD7A, and three (14.3%) of them showed THSD7A positivity. Seventy male patients were analyzable for THSD7A, and five (7.1%) of them showed THSD7A positivity. Non-tumor tissue of the lung was mainly negative for THSD7A. Representative images are shown in Figure 1. In the subgroup of female patients, THSD7A positivity was associated with poor overall survival (*p* = 0.027, Figure 2A). No significant association with overall survival was seen for the whole cohort and for the subgroup of male patients (*p* = 0.91 and *p* = 0.37, respectively, Figure 2B,C). THSD7A positivity was linked to the keratinizing subtype of LSCC (*p* = 0.049), but there was no significant association with clinicopathological parameters, neither for the whole analyzed population nor for the subgroups of females and males. The results are shown in detail in Table 1, Table 2 and Table 3.

A total of 91 (90.1%) tumors were analyzable for FAK-IHC. A total of 75 (82.4%) tumors showed a low expression, and a total of 16 (17.6%) tumors showed a high expression with a cytoplasmic staining pattern, respectively. Twenty-one female patients were analyzable for FAK, and four (19.0%) of them showed a high FAK expression. Seventy male patients were analyzable for FAK, and twelve (17.1%) of them showed a high FAK expression. Non-tumor tissue of the lung at best revealed a very low cytoplasmic FAK expression. Representative images are shown in Figure 3. 

In the subgroup of female patients, a high FAK expression was linked to poor overall survival (*p* = 0.058, Figure 4A). In the subgroup of male patients, a low FAK expression was linked to poor overall survival (*p* = 0.045, Figure 4B). No significant association was seen regarding the whole subgroup (*p* = 0.3, Figure 4C).

There was no significant association between FAK expression and clinicopathological parameters, neither for the whole analyzed population nor for the subgroups of females and males. The results are shown in detail in Table 1, Table 2 and Table 3.

### 2.1. Association between THSD7A Positivity and FAK Expression

A total of 91 (90.1%) tumors were analyzable for FAK-IHC as well as for THSD7A-IHC. There was no significant association between THSD7A positivity and FAK expression regarding the whole population and the subgroup of male patients, but THSD7A positivity showed a tendency towards high FAK-expression (*p* = 0.080) in female patients. The results are shown in Table 4.

### 2.2. Association of THSD7A Positivity with FGFR1 Gene Amplification

Only one THSD7A-positive case also showed an amplification in the FGFR1 gene. In female patients, none of the THSD7A-positive tumors showed a FGFR1-amplification (Table 1, Table 2 and Table 3).

### 2.3. Association of FAK Expression with FGFR1 Gene Amplification

No significant association could be shown between the expression level of FAK and the amplification status of FGFR1. However, only two (9.1%) FGFR1-amplified cases showed a high FAK-expression (Table 1, Table 2 and Table 3).

## 3. Discussion

Squamous cell carcinoma represents one of the most common histological subtypes of lung cancer. Nonetheless, there still is quite a lack of prognostic and predictive markers for this tumor entity. With respect to the poor prognosis of LSCC, novel predictive (molecular) markers and targets are desirable. 

In a recent study, we could show that THSD7A positivity is associated with high FAK expression in prostate cancer [5]. For that reason, we examined the role of THSD7A as a potential biomarker in squamous cell carcinomas of the lung and the association between THSD7A and FAK in this tumor entity.

Concerning the association between THSD7A expression and clinicopathological parameters in LSCC, data are missing so far. In this study, a total of eight (8.8%) tumors showed at least weak THSD7A positivity. Regarding the entire cohort, THSD7A positivity was linked to the keratinizing subtype of LSCC, but there was no significant association with any clinicopathological parameter, nor with the FAK expression level or FGFR1 gene amplification. The subdivision of the entire cohort into female and male patients, however, revealed very interesting correlations: THSD7A positivity was associated with poor overall survival and showed a tendency towards high FAK expression in the female cohort. No significant associations were seen in the male cohort in this regard.

We could not find a significant association between FAK expression and clinicopathological parameters or with the FGFR1 gene amplification status, neither for the whole analyzed population nor for the subgroups of females and males. These results are pretty much in the range of previously collected data, although, in contrast to others, we could find some interesting tendencies when dividing patients according to gender. In the subgroup of female patients, high FAK expression was linked to poor overall survival (*p* = 0.058), whereas in the subgroup of male patients, low FAK expression was linked to poor overall survival (*p* = 0.045).

Several groups analyzed FAK expression in lung cancer, though only a small number of studies deal with FAK expression in LSCC. Nan et al. report on an overexpression of FAK in premalignant lesions of the lung (squamous metaplasia and atypical adenomatous hyperplasia) using Western blot [51]. However, the number of analyzed samples is quite small (*n* = 13), and a statement concerning invasive tumors is missing. Wang et al. could not find a significant association between the activated/phosphorylated form of FAK and clinicopathological parameters in non-small cell lung cancer [38]. They reported a tendency towards a shorter survival time in the group with strong FAK expression, though this association did not reach significance. However, the analyzed cohort consisted of both squamous cell carcinomas and adenocarcinomas. Aboubakar et al. could show that the expression of both total FAK and activated FAK was significantly higher in lung cancer compared to normal lung tissue and was significantly higher in small-cell lung cancer compared to non-small-cell lung cancer. However, they could not find a significant association of the FAK expression status with recurrence-free and overall survival. It is also worth mentioning that only about one third of the analyzed NSCLC cohort consisted of LSCC [39]. Ji et al. also investigated FAK expression in different types of NSCLC [40]. They state that a high FAK expression (clone 4.47) is significantly correlated with tumor stage and lymph node metastasis as well as with poor overall survival. However, significant associations could now be demonstrated in the subgroup of LSCC.

Furthermore, others have shown that activation of FAK signaling is associated with EGFR-TKI resistance in NSCLC [52], although this is currently not of great importance for squamous cell carcinoma of the lung. 

Several studies indicate that THSD7A might at least play a role in the prognosis of different tumor types and that it possibly activates FAK-dependent signaling pathways [5,9,11,13,14,15,16,17,18]. The aim of this study was to examine the role of THSD7A as a potential biomarker in LSCC. In addition, we were interested in the impact of THSD7A positivity on the expression level of FAK in its unphosphorylated form. An earlier study also enabled us to check whether these two proteins are related to FGFR1 gene amplification. In this former study, it was shown that FGFR1 amplification was linked to better overall survival in female patients and to late tumor stages in LSCC. Interestingly, only one THSD7A-positive case also showed an amplification in the FGFR1 gene. None of the THSD7A-positive tumors in female patients showed an FGFR1 gene amplification. In contrast to FGFR1 amplification, THSD7A positivity was correlated with poor overall survival in female patients. FGFR1 is a promising therapeutic target and prognostic marker in different tumor types [50,53]. Assuming that the amplification of FGFR1 might describe a specific manner of tumor development in LSCC, it is likely that THSD7A positivity describes a different manner of tumor development.

Like our recently published data on prostate cancer [5], THSD7A positivity showed at least a tendency towards high FAK expression in female patients with LSCC. This again might be proof of the assumed activation of FAK-dependent signaling pathways by THSD7A. High FAK expression was correlated with poor overall survival in this subgroup. Surprisingly, high FAK expression was linked to better overall survival in male patients.

A definite weakness of this study is that we only analyzed a relatively small cohort, and the group of female patients was even smaller than the group of male patients. Nevertheless, the prognostic impact of THSD7A positivity in the female group is quite striking. Furthermore, this analysis was conducted as a pure IHC-study, for reasons of feasibility as well. Therefore, it is not possible to make clear statements about functionality.

To our knowledge, we are the first to report on the association of THSD7A with clinicopathological parameters and overall survival in LSCC. We are also the first who investigated the relationship between THSD7A positivity and FAK expression status in LSCC. Based on a former study, we were able to compare the expression status of THSD7A and FAK with FGFR1 gene amplification in LSCC.

## 4. Materials and Methods

### 4.1. Material

A previously described cohort of primary LSCC was brought into a tissue microarray format. The samples came from 101 patients who were surgically treated between 2006 and 2013 at the Department of Thoracic and Cardiovascular Surgery of the University Hospital of Saarland, Germany. Details are shown in Table 1.

### 4.2. Tissue Microarrays

TMA construction was performed using a manual tissue Arrayer according to the manufacturer’s directions (Manual Tissue Arrayer, AlphaMetrix Biotech, Rödermark, Germany). Tissue cylinders with a diameter of 1.0 mm each were punched out of selected paraffin-embedded tumor tissue blocks and were brought into empty “recipient” paraffin blocks. Sections from the TMA blocks measuring 4 µm were transferred to adhesion slides (Matsunami TOMO) and were used for immunohistochemistry (IHC). For a total of 19 cases, whole slide validation was carried out to increase the number of analyzable cases.

### 4.3. Immunohistochemistry

The TMA blocks were cut into 4 µm sections, transferred to adhesive slides (Matsunami TOMO) using a water bath (46 °C) and dried overnight at 37 °C. Staining was performed with Benchmark Ultra (Ventana Medical Systems) using primary antibody specific for THSD7A (rabbit polyclonal antibody, Sigma Aldrich, St. Louis, MO, USA; cat# HPA000923; dilution 1:150) and FAK (mouse monoclonal antibody (clone 4.47), Millipore; cat# 05-537; dilution 1:100).

Bound antibody was then visualized using the ultraView Universal Alkaline Phosphatase Red Detection (Roche, Basel, Switzerland) according to the manufacturer‘s directions. Heat-induced antigen retrieval at 97 °C was performed with CC2 buffer (Ventana/Roche, Basel, Switzerland) for 56 minutes (THSD7A) and with CC1 buffer (Ventana) for 64 minutes (FAK), respectively. 

For evaluation of THSD7A expression, the percentage of positive cells was estimated, and the staining intensity was recorded semiquantitatively as 0, 1+, 2+, or 3+ for each tissue sample. This was carried out by an experienced pathologist (P.R.S.). The staining results were categorized into the following four groups for statistical analysis: Tumors without any staining were considered negative. Tumors with 1+ staining in ≤70% or with 2+ staining in ≤30% of cells were considered weakly positive. Tumors with 1+ staining in >70%, with 2+ staining in >30% but ≤70%, and with 3+ staining in <30% of cells were considered moderately positive. Tumors with 2+ staining in >70% and with 3+ staining in ≥30% of cells were considered strongly positive. To better define THSD7A expression, we dichotomized THSD7A expression as negative (no staining in any tumor cell) and positive (at least 1+ staining in at least a few tumor cells).

For evaluation of FAK expression, the percentage of positive cells was estimated, and the staining intensity was recorded semiquantitatively as 0, 1+, 2+, or 3+ for each tissue sample. To better define FAK expression, we dichotomized FAK expression as low (tumors with 0 staining, 1+ staining, 2+ staining in ≤70% and with 3+ staining in ≤30% of cells) and high (tumors with 2+ staining in >70% of cells and 3+ staining in >30% of cells).

### 4.4. Statistics

Statistical analysis was performed using R (R Corporation 2021, R Foundation for Statistical Computing, Vienna, Austria). Pearson’s Chi-squared test (with Yate’s correction for continuity) was used for testing the null hypothesis of independence of two categorical variables. Kaplan–Meier curves and the log-rank test were used to test for differences in the survival time between the subgroups.

## 5. Conclusions

THSD7A positivity predicts poor overall survival in female patients with squamous cell carcinoma of the lung and shows a relation to high FAK expression and FGFR1 wild type in this subgroup. This might be proof of the assumed activation of FAK-dependent signaling pathways by THSD7A. As a membrane-associated protein, THSD7A might serve as a putative therapeutic target in LSCC therapy.

## Figures and Tables

**Figure 1 ijms-24-10639-f001:**
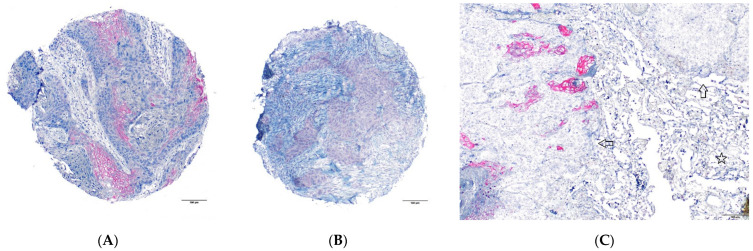
THSD7A shows membranous and cytoplasmic staining pattern: (**A**) tumor cells with membranous staining and (**B**) tumor cells with cytoplasmic staining; (**C**) whole slide (large section validation) showing tumor cells (arrows) with at least focal THSD7A positivity, while the adjacent non-tumor tissue (star) is THSD7A-negative.

**Figure 2 ijms-24-10639-f002:**
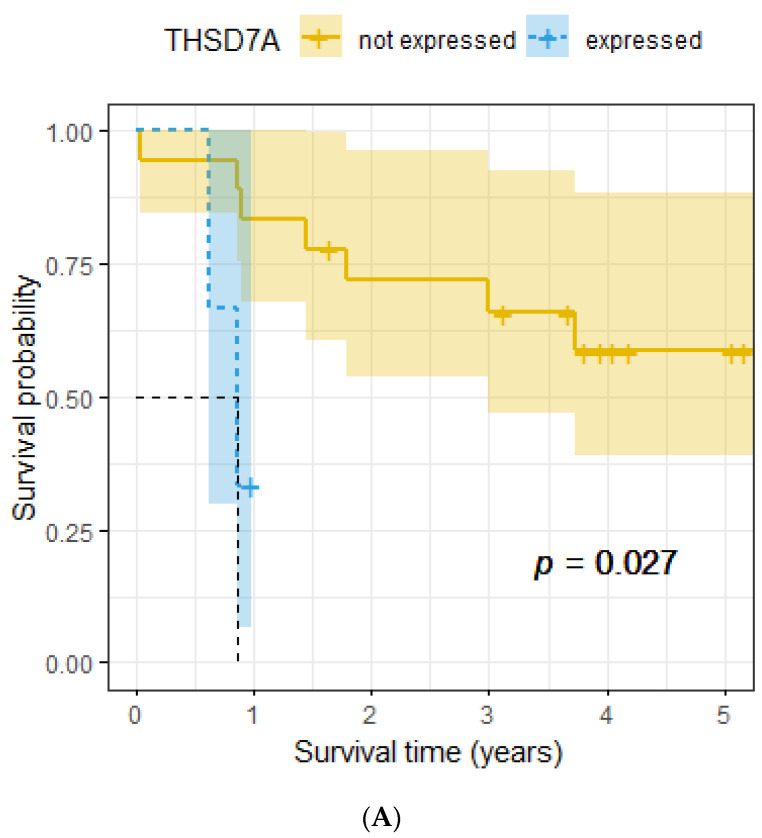
(**A**) Association between THSD7A positivity and overall survival, female subgroup. (**B**) Association between THSD7A positivity and overall survival, male subgroup. (**C**) Association between THSD7A positivity and overall survival, whole cohort.

**Figure 3 ijms-24-10639-f003:**
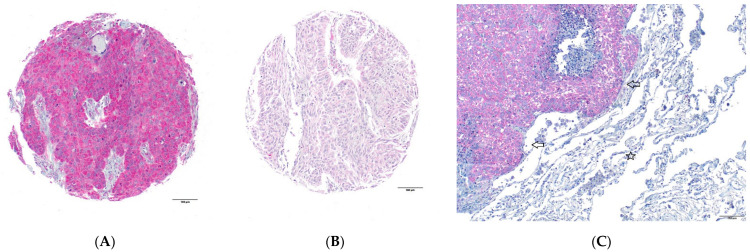
FAK shows cytoplasmic staining pattern: (**A**) tumor cells with high FAK expression and (**B**) tumor cells with low FAK expression; (**C**) whole slide (large section validation) showing tumor cells (arrows) with a high FAK expression, while adjacent non-tumor tissue (star) shows, at best, a very low FAK expression.

**Figure 4 ijms-24-10639-f004:**
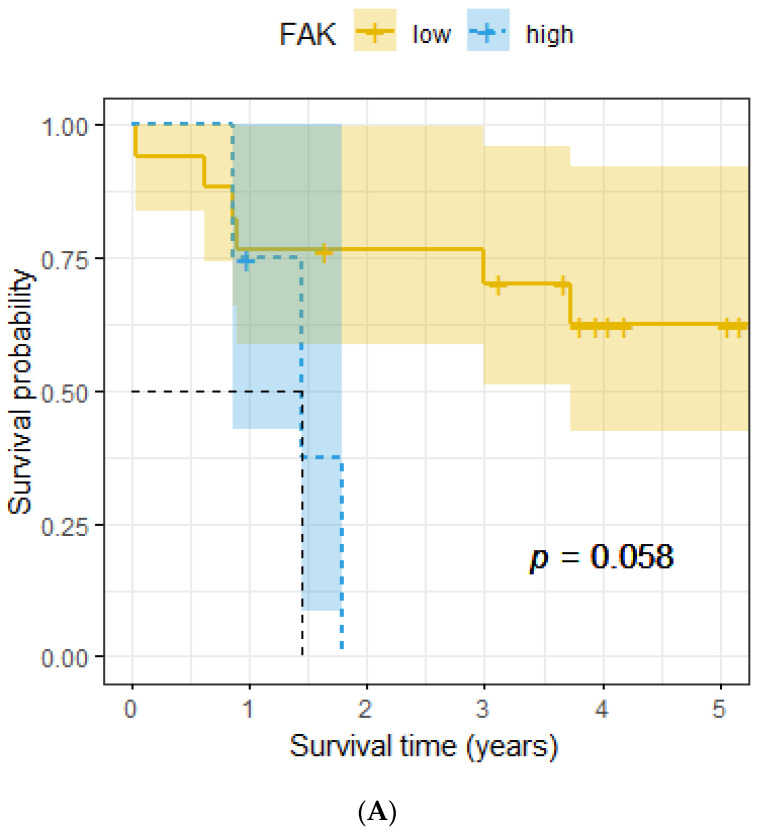
(**A**) Association between FAK expression and overall survival, female subgroup. (**B**) Association between FAK expression and overall survival, male subgroup. (**C**) Association between FAK expression and overall survival, whole cohort.

**Table 1 ijms-24-10639-t001:** Association of THSD7A expression and FAK expression with clinicopathological parameters and with FGFR1, whole cohort.

		THSD7A		FAK		
	Negative (*n* = 83)	Positive (*n* = 8)	*p*-Value	Low (*n* = 75)	High (*n* = 16)	*p*-Value
Sex			0.4			>0.9
female	18	3		17	4	
male	65	5		58	12	
Histologic type			0.049			0.090
non-keratinizing	55	2		44	13	
keratinizing	28	6		31	3	
Tumor stage			0.8			0.7
pT1	20	1		18	3	
pT2	33	3		30	6	
pT3	17	3		17	3	
pT4	13	1		10	4	
Nodal status			>0.9			0.9
pN0	50	5		45	10	
pN+	33	3		30	6	
Grading			0.5			>0.9
G1	0	0		0	0	
G2	40	5		37	8	
G3	43	3		38	8	
FGFR1 status			0.7			0.3
no amplification	62	7		55	14	
amplified	21	1		20	2	

**Table 2 ijms-24-10639-t002:** Association of THSD7A expression and FAK expression with clinicopathological parameters and with FGFR1, female group.

		THSD7A		FAK		
	Negative (*n* = 18)	Positive (*n* = 3)	*p*-Value	Low (*n* = 17)	High (*n* = 4)	*p*-Value
Histologic type			>0.9			>0.9
non-keratinizing	13	2		12	3	
keratinizing	5	1		5	1	
Tumor stage			0.7			0.9
pT1	3	0		3	0	
pT2	9	1		7	3	
pT3	4	2		5	1	
pT4	2	0		2	0	
Nodal status			0.2			>0.9
pN0	13	1		11	3	
pN+	5	2		6	1	
Grading			>0.9			0.6
G1	0	0		0	0	
G2	10	2		9	3	
G3	8	1		8	1	
FGFR1 status			0.5			0.5
no amplification	13	3		12	4	
amplified	5	0		5	0	

**Table 3 ijms-24-10639-t003:** Association of THSD7A expression and FAK expression with clinicopathological parameters and with FGFR1, males.

		THSD7A		FAK		
	Negative (*n* = 65)	Positive (*n* = 5)	*p*-Value	Low (*n* = 58)	High (*n* = 12)	*p*-Value
Histologic type			0.008			0.11
non-keratinizing	42	0		32	10	
keratinizing	23	5		26	2	
Tumor stage			>0.9			0.5
pT1	17	1		15	3	
pT2	24	2		23	3	
pT3	13	1		12	2	
pT4	11	1		8	4	
Nodal status			0.4			>0.9
pN0	37	4		34	7	
pN+	28	1		24	5	
Grading			0.7			0.8
G1	0	0		0	0	
G2	30	3		28	5	
G3	35	2		30	7	
FGFR1 status			>0.9			0.7
no amplification	49	4		43	10	
amplified	16	1		15	2	

**Table 4 ijms-24-10639-t004:** (**A**) Association between THSD7A positivity and FAK expression, female subgroup. (**B**) Association between THSD7A positivity and FAK expression, male subgroup. (**C**) Association between THSD7A positivity and FAK expression, whole cohort.

(A)				
		**THSD7A**		
	**Negative, *n* = 18**		**Positive, *n* = 3**	***p*-Value**
				0.080
FAK				
Low level	16		1	
High level	2		2	
**(B)**				
		**THSD7A**		
	**Negative, *n* = 65**		**Positive, *n* = 5**	***p*-Value**
				>0.9
FAK				
Low level	54		4	
High level	11		1	
**(C)**				
		**THSD7A**		
	**Negative, *n* = 83**		**Positive, *n* = 8**	***p*-Value**
				0.14
FAK				
Low level	70		5	
High level	13		3	

## Data Availability

The datasets used and analyzed in this paper are available from the corresponding author on reasonable request.

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
