# Peer review of "THSD7A Positivity Predicts Poor Survival and Is Linked to High FAK Expression and FGFR1-Wildtype in Female Patients with Squamous Cell Carcinoma of the Lung"

_ijms, 2023, doi:10.3390/ijms241310639_

Round 1

Reviewer 1 Report

The authors point out that although lung cancer is the leading cause of cancer-related deaths in the Western world, prognostic and predictive markers are still mainly missing for squamous cell carcinoma of the lung (LSCC).

Thrombospondin Type-1 Domain 37 containing 7A (THSD7A) is a promising marker for other tumors, but its role has never been investigated concerning lung cancer. In a recent study, the authors showed that positivity of THSD7A is associated with high expression of Focal Adhesion Kinase 38 (FAK) and with adverse clinicopathological parameters in prostate cancer.

However, data on the association between THSD7A and FAK expression and clinicopathological parameters in LSCC are missing.

Here authors aim to fill the gap. In detail, they conducted this study to investigate the role of THSD7A as a potential biomarker in LSCC and to see whether THSD7A positivity impacts the expression of FAK in its unphosphorylated form. The authors found a correlation between THSD7A and FAK expression and LSCC.

The aim of the paper and the results are clear. The title is informative and relevant. The research question is clearly outlined.

The introduction provides sufficient background for the role of THSD7A and FAK but not for FGFR1. It is not clear the link between THSD7A, FAK, and FGFR1 and between FGFR1 and lung cancer.

Revisions

There are no explained links between FGFR1 and THSD7A, and FAK. Please, try better to define the role of FGFR1 in lung cancer. In the tables, it seems to be considered a clinicopathological parameter.

Line 65 Authors report that the mutation frequency for FGFR1 is 22%. Please, explain where this number comes from since it is mentioned here but not in the result section.

Fig 1 and 3. Are Figs A and B part of Fig C? In Fig C, please, indicate healthy and nonhealthy tissue and adjust the magnification bar to make it readable.

Fig 2 and 4. Please, describe the analysis method used and what the p-value represents.

Line 256 Please, specify how the staining intensity was recorded and measured.

Line 250 Please, specify the source of the FAK antibody (mouse?)

Statistics. Please, specify which method is used for survival analysis.

Weaknesses.

The small number of analyzed samples. However, the authors consider this limit in the discussion. The paper can be viewed as a starting point to examine THSD7A and FAK’s role in LSCC.

Author Response

We thank Reviewer 1 for his comments and tried to answer the requests in an appropriate way.

There are no explained links between FGFR1 and THSD7A, and FAK. Please, try better to define the role of FGFR1 in lung cancer. In the tables, it seems to be considered a clinicopathological parameter.

Reviewer 1 asks us to better define the role of FGFR1 in lung cancer and to draw possible connections with THSD7A and FAK, resepectively. We provided more information on FGFR1 as a potential biomarker and tried to point out connections with FAK and THSD7A in lung cancer (Line 68-79).

Reviewer 1 is right that the arrangement of our tables is misleading with reference to FGFR1. FGFR1 is not an established clinicopathological parameter in lung cancer. We chose this arrangement because we wanted to demonstrate that the data on FGFR1 was previously collected. To avoid misunderstandings, we changed the headings of Table 1, Table 2 and Table 3.

Line 65 Authors report that the mutation frequency for FGFR1 is 22%. Please, explain where this number comes from since it is mentioned here but not in the result section.

This data was generated in a previous study. We explain that in Line 62-67 and provide the according reference. To avoid misunderstandings, we provide the according reference again in Line 67.

Fig 1 and 3. Are Figs A and B part of Fig C? In Fig C, please, indicate healthy and nonhealthy tissue and adjust the magnification bar to make it readable.

Figures A and B come from the constructed tissue microarry. Figure C comes from a large section validation and serves to show the difference between tumor tissue and non-tumor tissue. We modified Figure 1C and Figure 3C and made changes in the legends to figures.

Fig 2 and 4. Please, describe the analysis method used and what the p-value represents.

Statistics. Please, specify which method is used for survival analysis.

We added the requested information on the analysis method and p-value to the Statistics section (Line 287-290).

Line 256 Please, specify how the staining intensity was recorded and measured.

This was carried out by an experienced pathologist (P.R.S.). This information was added (Line 272).

Line 250 Please, specify the source of the FAK antibody (mouse?)

It is a mouse monoclonal antibody. This information was added (Line 265).

Reviewer 2 Report

The article is original and very interesting.

The conclusions are consistent with the evidence and arguments presented and authors have accurately identified the limitations of the study.

The references are appropriate, including some very relevant author’s previous experience in the field, but should be edited according to Journals Instructions for authors. According to doi, ref 7 and 9 are identical

I suggest some minor corrections.

When you cite many references for the same paragraph, follow the Journals Instructions for authors.

Eg. line 46 you may cite [20-26]

line 49 you may cite [27-29]

line 52 you may cite [5, 11-12]

line 55 you may cite [30-33]

line 56 you may cite [34-38]

line 59 you may cite [39-43]

line 202, put references in numerical order and you may cite [5,10,12,14-17, 19, 50]

Table 4 a,b, c- could be joined in a single one

Author Response

We thank Reviewer 2 for his comments and tried to answer the requests in an appropriate way.

The article is original and very interesting.

The conclusions are consistent with the evidence and arguments presented and authors have accurately identified the limitations of the study.

The references are appropriate, including some very relevant author’s previous experience in the field, but should be edited according to Journals Instructions for authors. According to doi, ref 7 and 9 are identical

I suggest some minor corrections.

When you cite many references for the same paragraph, follow the Journals Instructions for authors.

Eg. line 46 you may cite [20-26]

line 49 you may cite [27-29]

line 52 you may cite [5, 11-12]

line 55 you may cite [30-33]

line 56 you may cite [34-38]

line 59 you may cite [39-43]

line 202, put references in numerical order and you may cite [5,10,12,14-17, 19, 50]

The references were edited.

Table 4 a, b, c- could be joined in a single one

We agree that the three tables could be joined in a single. We would leave this decision to the editors.

Reviewer 3 Report

Dear Editor and Authors,

I would like to thank you for asking me to review this manuscript titled “THSD7A positivity predicts poor survival and is linked to high FAK expression and FGFR1-wildtype in female patients with squamous cell carcinoma of the lung” by Dr. Flockerzi and colleagues from the Saarland University Medical Center in Homburg, Germany.

In this study the authors, who have extensive experience on this specific marker, investigate the role of THSD7A as a potential biomarker in squamous cell carcinoma of the lung and to also establish whether THSD7A positivity is related with the expression of FAK. They utilized immunohistochemistry using tissue microarrays to study expression in 91 patient samples/tumours. The analysis demonstrated that female patients had a higher (14.3%) THSD7A expression compared to men and this also translated to lower overall survival. Similarly FAK expression in women was higher and survival was also affected.

This is a well conducted and straight forward study. It utilized solid methodology and a sound analysis. The sample of patients analyzed can be considered smallish but should be able to give statistically meaningful results. The manuscript is well written and well illustrated/reported. The language written is clear and understandable with some minor spelling errors.

Overall, I am content to recommend the publication of this work. Kind regards to all.

Language is good.

Author Response

We thank Reviewer 3 for his comments

This is a well conducted and straight forward study. It utilized solid methodology and a sound analysis. The sample of patients analyzed can be considered smallish but should be able to give statistically meaningful results. The manuscript is well written and well illustrated/reported. The language written is clear and understandable with some minor spelling errors.

Overall, I am content to recommend the publication of this work. Kind regards to all.

We fixed some spelling errors.
